# Hypoxia-Inducible Factor 1 and Mitochondria: An Intimate Connection

**DOI:** 10.3390/biom13010050

**Published:** 2022-12-27

**Authors:** Xiaochen Huang, Li Zhao, Ruiyun Peng

**Affiliations:** Beijing Institute of Radiation Medicine, Beijing 100850, China

**Keywords:** HIF-1, HIF-1α, mitochondria

## Abstract

The general objective of the review is to explain the interaction between HIF-1 and mitochondria. On the one hand, this review describes the effects of HIF-1 on mitochondrial structure, including quantity, distribution, and morphology, as well as on mitochondrial metabolism and respiratory function. On the other hand, various factors, including mitochondrial activation of enzymes, the respiratory chain, complex and decoupling proteins, affect the stability and activity of HIF-1. It is possible to develop future molecular therapeutic interventions by understanding the interrelationships between HIF-1 and mitochondria.

## 1. Introduction

As a mediator of intercellular hypoxia, hypoxia-induced factor 1 (HIF-1) plays a critical role; this is upholding the body’s homeostasis by activating multiple downstream target genes [1]. Mitochondria are the core of cell metabolism, providing enough energy and substances for cells, and are one of the most important oxygen-consuming organelles in cells [2]. Based on existing research, several findings have supported that HIF-1 regulates mitochondrial respiration, oxidative stress, and several other important processes, including those involved in HIF-1 target genes [3,4,5]. The cAMP response element-binding protein (CREB) is a ubiquitous transcription factor in the higher eukaryotes [6]. Together, HIF-1α and CREB regulate the expression of genes encoding proteins with roles in mitochondrial function, including members of the electron transport complex involved in ATP production [7]. Nevertheless, growing evidence from recent studies indicates that there is a mitochondria-triggered HIF-1 activation under various conditions such as hypoxia [8,9,10]. As a result, HIF-1 and mitochondria may interact and take a vital regulatory part in physiological and pathological processes in vivo; research into their relationship has, therefore, gradually gained traction.

This review summarizes the most recent research status of HIF-1 and mitochondrial interactions, with the goal of providing a reference for the regulation and intervention of HIF-1 and the mitochondrial role.

## 2. HIF-1

Semenza and his coworkers [11] discovered HIF-1 in 1992 during a genetic study on erythropoietin (EPO). HIF-1 belongs to the basic helix–loop–helix Per-Arnt-Sim (bHLH-PAS) transcription factor superfamily. The heterodimer consists of an oxygen-sensitive α-subunit (HIF-1α), as well as a constitutively expressed β-subunit (HIF-1β) [12]. The α subunit is widely distributed, tightly regulated by oxygen concentration, and contributes greatly to the regulation of hypoxia. As a structural subunit, the β subunit is insensitive to oxygen concentration, and its expression in cells is unaffected by normoxia or hypoxia.

The activation of HIF-1 is primarily dependent on its expression and activity [13]. During normoxia, when oxygen levels are not restricted, a family of prolyl-4-hydroxylases, known as PHD enzymes, regulates the stability and abundance of the HIF-1α protein [14]. The transcriptional activity of the HIF complex is also regulated by a transcription inhibitor called the Factor Inhibiting HIF-1 (FIH-1). Its name is derived from the initial description of the protein’s function, which is to suppress HIF-1 action by binding to HIF-1α. Aside from hydroxylation, HIF-1α undergoes several post-translational modifications, including acetylation, phosphorylation, nitrosylation, and SUMOylation (SUMO) [15,16]. Under hypoxic conditions, the function of hydroxylase is inhibited, and the stably expressed HIF-1 combines with HRE to promote the transcription and expression of target genes [17]. HIF-1 can be activated by various extracellular stimuli, and its nuclear translocation can transactivate more than 1000 genes. Consequently, HIF-1 is engaged in numerous biological processes, such as the regulation of mitochondrial function, energy metabolism, cell differentiation, and crucial life processes such as angiogenesis, inflammation, and immunological modulation. Considering the substantial effects of HIF-1, which participates in numerous essential physiological processes such as cardiovascular generation, cartilage development, neural embryo formation, and tumor growth, it is closely associated with a number of human pathological processes [18].

## 3. Mitochondria

In most eukaryotes, mitochondria are ubiquitous intracellular organelles that generate cellular energy. Mitochondria have two membranes surrounding them, a mitochondrial outer membrane (OMM) and an inner mitochondrial membrane (IMM); these separate the organelle into two parts, the matrix, and the intermembrane space (IMS) [19]. The OMM is smooth and has the function of organelle boundary membrane. There are large porins on it, which can allow molecules with a relative molecular mass of approximately 5 kDa to pass through. There are also lipid-synthesizing enzymes on the OMM, in addition to lipids that are converted into enzymes that can be further metabolized in the matrix. The IMM develops cristae, which are densely packed invaginations in the matrix. The IMM allows only water, oxygen (O_2_), and carbon dioxide (CO_2_) to pass freely. Large molecules and ions need a special transport system to pass through the inner membrane as membrane translocase. The IMM contains respiratory chain enzymes and adenosine triphosphatase (ATPase) complexes, which undertake more biochemical reactions. IMM also plays a crucial role in energy conversion. All enzymes needed for the tricarboxylic acid cycle (TCA) are housed in the matrix, which is located in the center of the mitochondria. Furthermore, it consists of filaments, as well as dense granular substances with high electron density (including Ca^2+^, Mg^2+^, and Zn^2+^ plasma), and has a complete set of transcription and translation systems, namely the mitochondrial genome and mitochondrial DNA (mtDNA).

Mitochondria are the location of the oxidative metabolism in eukaryotes. Their main functions are providing energy, oxidizing, and converting various energy substances such as carbohydrates, fats, and proteins. This process of generating energy by oxidizing various substances is called aerobic respiration, which is the basis of cell life activities [19]. In addition, mitochondria are critical in maintaining calcium homeostasis, regulating membrane potential, controlling apoptosis, innate immunity production, β-oxidation, proteostasis, lipid synthesis, the urea cycle, nucleotide metabolism and maintaining cellular pH balance [20].

Moreover, mitochondria evolved from free-living bacteria and are involved in the origin of eukaryotic cells by a process referred to as endosymbiosis. By means of the respiratory chain, mitochondria produce adenosine triphosphate (ATP) [19,21]. As a result of their role in the production of energy, mitochondria are sometimes referred to the “powerhouse of the cell” [22]. As well as being a signaling organelle, a variety of physiological functions are also performed by mitochondria, such as apoptosis, heme, and the synthesis of iron-sulfur clusters. Due to the very high iron content of the mitochondria, iron is known to play a major role in mitochondrial function; this includes producing iron-sulfur (Fe-S) clusters, and in the addition of heme synthesis. In the mitochondrion, there is a well-documented role for a pool of redox iron in free radicals (reactive oxygen species, ROS) mitochondrial accumulation (mitoROS). Due to the sensitivity of mitochondria, changes in the internal and external environment of cells can cause abnormal mitochondrial structure and function, thereby affecting various functions of cells, and even causing cell death. Therefore, mitochondria can be used as indicators for disease diagnosis and the determination of environmental factors.

## 4. Regulation of HIF-1 on Mitochondrial Structure and Function

### 4.1. Effects of HIF-1 on Mitochondrial Morphology and Structure

#### 4.1.1. HIF-1 and Mitochondrial Population

The population of mitochondria is mainly related to the type of cells and energy demand. Generally speaking, there are more mitochondria in the cells with vigorous metabolism and more energy requirements. In the same type of cells, the number of mitochondria is relatively stable, while the mitochondrial population will be changed with the fluctuation of cell function. Thus, mitochondrial number is an important cue for cellular homeostasis.

The population of mitochondria is closely related to multiple mitochondrial production and degradation processes, including mitochondrial biogenesis, autophagy, and mitochondrial dynamics [23,24]. Numerous studies have found that HIF-1 mediates mitochondrial biogenesis, mitophagy, and mitochondrial dynamics to regulate mitochondrial population.

Mitochondrial biogenesis can regulate mitochondrial number by the process which existing mitochondria generate new mitochondria to maintain mitochondrial homeostasis. Previous research has suggested that HIF-1 reduced mitochondrial number by inhibiting the PPAR-γ coactivator-1 (PGC-1) family members PGC-1α and PGC-1β, which are the essential transcription factors of mitochondrial biogenesis. Slot et al. [25] demonstrated that hypoxia inhibited PPAR/PGC-1α signaling and HIF-1α-mediated mitochondrial component expression in C2C12 cells. Lu et al. [26] demonstrated that in an in vitro model of mice genioglossus myoblast, the HIF-1α protein inhibited adenosine 5′-monophosphate-activated protein kinase (AMPK) under hypoxic conditions. Downregulation of HIF-1α increased the expression of the myogenin, PGC-1β and pAMPKα1, promoted the differentiation of myoblasts, and protected mitochondrial integrity. Inhibition of the AMPK pathway inhibited mitochondrial biogenesis, decreased the level of PGC-1β, and increased apoptosis. Zhang et al. [27] found that HIF-1 inhibited cancer-myc (C-MYC) activity in renal carcinoma cells lacking von Hippel–Lindau tumor suppressors (VHL). C-MYC activity was inhibited by HIF-1, which mediated these effects. As a result of the C-MYC-dependent reduction in PGC-1β expression, mitochondrial numbers were reduced, resulting in a low energy requirement for proliferation and survival of renal carcinoma cells.

Autophagy in mitochondria is the process of catabolizing cytoplasmic organelles to eliminate defective structures or to recycle them for future use. It plays key physiological cellular functions and is an important mechanism for controlling mitochondrial number. Recent studies demonstrated that Bcl-2/adenovirus E1B 19kDa-interacting protein 3 (BNIP3), a known HIF-1 target gene that had been implicated in autophagy, could reduce mitochondrial population [28]. To ensure the adaptive metabolic responses of cells in hypoxia, this mechanism required the HIF-1-dependent expression of BNIP3 and constitutive expression of Beclin-1 and Atg5, which lowered mitochondrial numbers and prevented a increase in ROS [29]. However, Fu et al. [30] verified that HIF-1α-BNIP3-mediated mitophagy functioned as a protective mechanism for acute kidney damage by inhibiting apoptosis and ROS production. In addition, Madhu et al. [31] verified hypoxia regulated mitophagy, through HIF-1α, by controlling BNIP3 translocation to mitochondria.

Mitochondrial dynamics refers to the dynamic process of mitochondrial fusion and fission that is controlled by GTPases. The key regulatory proteins included OMM fusion mitochondrial fusion protein 1/2 (Mfn1/2), while IMM fusion exogenous hydrogen sulfide regulates optic atrophy 1 (OPA1)-mediated mitochondrial fission dynamin-related protein 1 (DRP1) [32]. HIF-1 regulated the above-mentioned key proteins of mitochondrial fusion and fission, participated in mitochondrial dynamics, and affected the number of mitochondria. In the process of mitochondrial fission, on the one hand, most research indicated that HIF-1α positively regulated DRP-1 to promote mitochondrial fission. According to Marsboom et al. [33], HIF-1 activation caused mitochondrial fission in human models of pulmonary arterial hypertension (PAH) by phosphorylation of DRP1 at serine 616 via cyclin B1/CDK1-dependent inhibition, whereas in normal pulmonary smooth muscle cells (PASMC), DRP1-mediated fission was caused by HIF-1 activation by cobalt chloride (CoCl2) or desferrioxamine. Therefore, HIF-1α inhibition could reduce PASMC proliferation by preventing fission through DRP1 inhibition. As observed by Wan et al. [34], hypoxia increased Drp1 transcription and expression in glioblastoma U251 cells, while stimulating mitochondrial fission. Echinomycin’s inhibition of HIF-1α prevented the hypoxia-induced expression of Drp1. In particular, the Drp1 inhibitor Mdivi-1 effectively attenuated hypoxia-induced mitochondrial fission and migration in U251 cells. Additionally, it was also reported that HIF-1α negatively regulated DRP-1 and Mfn2 to reduce mitochondrial fission. According to Pan et al. [35], the HIF/miR125a/Mfn2 pathways regulated mitochondrial fission, which influenced PANC1 cell survival, growth, metabolism, and migration, posing a potential treatment for pancreatic cancer (PC). Jiang et al. [36] verified that HIF-1α exerted a protective effect against tubular injury in a mouse model of diabetic nephropathy (DN), which was mediated via modification of mitochondrial dynamics via HO-1 overexpression. In studies of mitochondrial fusion, it was found that HIF-1α affected Mfn1/2 to change mitochondrial fusion. Chiche et al. [37] revealed that when exposed to Hypoxia over long periods of time, some cancer cells were capable of evading apoptosis by initiating mitochondrial fusion, targeting BNIP3 and BNIP3L in mitochondrial membranes, and updating the expression of Mfn1. Therefore, these cells had a selective growth advantage; this was a process dependent on the regulation of HIF-1α. Chen et al. [38] found that in both PASMC and PAH rats, HIF-1α regulated mitochondrial dynamics during hypoxia-induced pulmonary vasculature remodeling by directly lowering Drp1 expression and increasing the expression of Mfn2 (Figure 1). 

#### 4.1.2. HIF-1 and Mitochondrial Distribution

Mitochondria are distributed in most cells and are dynamically adjusted to the energy demands of different cell and environmental types. The transport of mitochondria can be divided into two categories: anterograde movement from the nucleus to the periphery and retrograde movement from the periphery to the nucleus. Hypoxia lead to retrograde redistribution to the nucleus. Mitochondria with normal membrane potential exhibit high levels of prograde movements. On the contrary, when the membrane potential and ATP synthesis activity are disturbed, mitochondria show high levels of retrograde movements [39].

HIF-1 could alter intracellular mitochondria distribution by regulating mitochondrial motility. Li et al. [40] demonstrated that the mitochondrial movement regulator (HUMMR) was upregulated by hypoxia, markedly induced by HIF-1α, and mitochondria transport was skewed in the anterograde direction for rational distribution throughout the neuron. Furthermore, HUMMR could rescue the mitochondrial content in axons dependent on HIF-1α. Either long-term hypoxia or increased expression of the protein coiled-coil helix domain-containing protein 4(CHCHD4) by U2OS tumor cells in normoxia resulted in an accumulation of mitochondria perinuclearly, which was dependent upon HIF-1α activation. A key feature of the cellular response to hypoxia was the intracellular distribution of the mitochondrial network, which contributed to hypoxic signaling via HIF activation and was regulated by cross-talk between CHCHD4 and HIF-1α. As part of the cellular response to hypoxia, the mitochondrial network was distributed intracellularly, which contributed to hypoxic signaling via HIF activation, and was regulated through cross-talk between CHCHD4 and HIF-1α [41]. Gentillon et al. [42] found that by combining HIF-1α inhibition with metabolic regulators treatment, there was an increase in mitochondrial content in human-induced pluripotent stem cells (hiPSC-CMs), as an increase in cells with mitochondrial distribution throughout, which enhanced the metabolic maturation.

#### 4.1.3. HIF-1 and Mitochondrial Structure

In normal physiological conditions, mitochondria are oblong under an electron microscope, with complete inner and outer membranes, cristae and matrix, and a clear intermembrane space. Mitochondria are connected in the cell, presenting a three-dimensional tubular network structure. Mitochondrial morphology is the structural basis for its function.

HIF-1 regulated mitochondrial morphology, such as size, shape, and structure, which might underlie functional changes or might be secondary to functional regulation [43,44]. Chiche et al. [37] incubated tumor cells in hypoxia, such as LS174Tr cells and renal cells carcinoma 786-Ocells, which had unusually enlarged mitochondria. Induced by tetracycline (Tet) of HIF-1α or the mutation of VHL, it was constitutive for stable HIF-1α protein through a HIF-1-dependent mechanism. Chen et al. [45] treated rat mesangial cells (RMC) with bicalutamide (Bic) to up-regulate the expression of HIF-1α. In the majority of the mitochondria, cristae were lacking and the mitochondria were moderately or severely swollen. It also indicated the impairment of mitochondrial metabolic function. Chen et al. [38] discovered that HIF-1α regulated Drp1 expression directly in hypoxia-induced pulmonary vascular remodeling.

### 4.2. Effects of HIF-1 on Mitochondrial Function

#### 4.2.1. HIF-1 and Mitochondrial Respiratory Chain

The mitochondrial matrix is the site of the oxidation of respiratory substrates; this reaction results in the production of nicotinamide adenine dinucleotide (NADH) and flavin adenine dinucleotide (FADH2). With the transfer of the protons and electrons from O_2_ to H_2_O, H_2_O is generated, passing through a series of hydrogen carriers and electron carriers in the IMM. The electron transfer chain (ETC), composed of carriers, is also called the respiratory chain for the direct relationship with respiration.

HIF-1 regulated the efficiency of the respiratory chain by acting on different mitochondrial respiratory chain complexes. Regueira et al. [46] used tumor necrosis factor-α (TNF-α) and CoCl_2_ to increase HIF-1α expression in the human hepatoma cell line HepG2, and found that mitochondrial oxygen consumption was decreased in complexes I, II, and IV-dependent mitochondria. Chetomin (CTM) prevented the decrease in cellular oxygen consumption, indicating that as a response to CoCl_2_ and TNF-α, HIF-1 controlled mitochondrial respiration. Fukuda et al. [47] demonstrated that in mammalian cells, HIF-1 reciprocally controlled cytochrome c oxidase 4 (COX4) subunit expression under circumstances of low O_2_ availability by promoting transcription of the genes encoding COX4-2 and LON, a mitochondrial protease necessary for COX1-1 breakdown. The change on COX4 subunit expression affected COX activity, ATP synthesis, and oxygen consumption. Douiev et al. [48] demonstrated that in the COX4-1-deficient human foreskin fibroblasts (HFF) cells, COX4-2 levels were elevated with a concomitant HIF-1α stabilization, nuclear localization and upregulation of the hypoxia and glycolysis pathways. COX4-2 and HIF-1α in normoxia were also upregulated as a compensatory mechanism of COX4-1 deficiency. Owing to ROS in cells, mainly coming from the electron leakage and damaging the mitochondrial respiratory chain, the effect of HIF-1 on ROS is often regarded as acting on the respiratory chain, which mainly affected ROS production by directly targeting the mitochondrial respiratory chain complex. HIF-1, which in hypoxic mammalian cells was negatively regulated by O_2_-dependent hydroxylation, increased the transcription of the COX4I2 and LON genes. COX4-2 mRNA and protein production was increased, while COX1 proteolysis was increased. Therefore, ROS production at toxic levels was prevented, and the respiratory efficiency of hypoxic cells was enhanced [47]. It was hypothesized that HIF-1 served as a ROS sensor to limit the excessive generation of mitochondrial ROS in response to cytokine stimulation [49] (Figure 2).

#### 4.2.2. HIF-1 and TCA Cycle

In the mitochondrial matrix, the TCA cycle is completed and the intermembrane space, there is a common metabolic pathway and liaison mechanism for the complete oxidation of sugar, fat, and protein in the body. Among them, carbon is mainly derived from three metabolites, namely glucose, fatty acid and glutamine [50]. Amides, glucose, and fatty acids are metabolized and decomposed into H_2_O and CO_2_ in the TCA cycle. Part of the metabolites of glutamine enters the TCA cycle. Then it is decomposed into succinyl-CoA undergo 2-oxoglutaric acid (2-OG), which is subsequently utilized by oxidation [51].

Previous studies have found that HIF-1 affected TCA cycle efficiency by acting on metabolites. HIF-1 positively regulated lactate dehydrogenase A (LDHA) expression and promoted the conversion of pyruvate to lactic acid in hypoxic conditions. Meanwhile, with the raise of 3-phosphoinositide dependent protein kinase 1 (PDK1) and suppression of pyruvate into acetyl coenzyme A, the content of acetyl coenzyme A into the TCA cycle was reduced [52,53]. HIF-1 promoted lactate excretion to maintain optimal cytoplasmic pH by upregulating the plasma membrane transporter monocarboxylate transporters 4 (MCT4), which stimulated lactic acid excretion, avoided the competitive inhibition of LDHA due to excessive lactic acid, and then accelerated the TCA cycle [54] (Figure 2).

#### 4.2.3. HIF-1 and Oxidative Phosphorylation

Through the oxidative phosphorylation (OXPHOS) process, mitochondria convert the energy generated by the oxidation of substrates into ATP that can be available to cells. Under hypoxic conditions, HIF signaling promoted anaerobic respiration to produce ATP and downregulated OXPHOS to reduce cellular oxygen-dependent energy requirements [55]. Liu et al. [56] demonstrated that, as a result of the nutritional deficiencies in U251 cells, HIF-1α protein expression was decreased, resulting in higher levels of C-MYC, which increased nuclear respiratory factor 1 (NRF1) and mitochondrial transcription factor A (TFAM) expression; this in turn, enhanced OXPHOS activity. During carcinogenesis, metabolic reprogramming might provide some new therapeutic clues for traditional cancer treatments due to changes in the microenvironment (Figure 2).

#### 4.2.4. HIF-1 and Mitochondrial Membrane Potential

The mitochondrial membrane potential (MMP) is one of the main mechanisms that ensures the normal physiological functions of mitochondria. The decrease in ΔΨm is a hallmark event of early apoptosis. Studies have suggested that HIF-1 negatively regulated ΔΨm. Watanabe et al. [57] demonstrated that UCP3 modulated CTP in response to hypoxia and was associated with a decrease in Bcl2 and BCL_xL_ expression in the cytoplasm, and an increase in baseline cytochrome c concentrations. Sasabe et al. [58] found that in the human oral squamous carcinoma cell (OSCC) lines, the overexpression of HIF-1α blocked the ΔΨm reduction and the cytosolic accumulation of cytochrome c resulted in the activation of caspase-9 and caspase-3. Multiple investigations have demonstrated that HIF-1 was a powerful apoptosis inhibitor.

## 5. The Effects of Mitochondria on HIF-1 Activity

Mitochondria are sensitive and changeable organelles, and their perception of the cellular microenvironment is more subtle than other organelles. Mitochondria act as oxygen-sensing organelles that transmit signals directly or indirectly to HIF-1. In addition, they contribute to the regulation of cellular redox, ion homeostasis, and energy production [57]. Although the important role of HIF-1 in the regulation of oxygen homeostasis has been recognized, HIF-1 accumulation and stability remains unclear in the absence of mitochondria. Most previous studies believe that ROS, generated by mitochondria, is the key factor in triggering HIF-1 activity, but with the deepening of research, more results have shown that various functions of mitochondria are closely related to HIF-1 (The mitochondrial roles and regulation of HIF-1 activity and stability are shown in Table 1).

### 5.1. Mitochondrial Metabolic Enzymes and HIF-1 Activity

Mitochondria are known to be rich in enzymes, with more than 120 species, of which oxidoreductase accounts for approximately 37%, ligase accounts for 10%, and hydrolase accounts for less than 9%. The most complex types of enzymes are in the IMM and matrix. They are involved in respiratory chain oxidation, ATP generation, TCA catalyzation, fatty acids and pyruvate oxidization. Therefore, mitochondrial metabolic enzymes are necessary to maintaining vitality [90].

Previous studies have suggested that various mitochondrial metabolic enzyme activities could affect the accumulation of HIF-1, mainly by activating its regulatory pathway or by inhibiting its degradation pathway. Selak et al. [59] demonstrated that TCA disorders led to the down-regulation of succinate dehydrogenase (SDH) and fumarate hydratase (FH) activities, which resulted in the accumulation of succinate, PHD activity inhibition, and HIF-1α induction. Fuhrmann et al. [60] treated THP-1 cells with the SDH inhibitor atpenin A5 (AA5), which provoked HIF-1α stabilization and concomitated HIF-1 target gene activation, with the macrophage phenotype shifting to classical cell activation. Liu et al. [61] investigated the effects in lipopolysaccharides (LPS)-stimulated mouse bone marrow-derived macrophages (BMDMs). Tanshinone IIA (Tan-IIA) suppressed mitochondrial reactive oxygen species production, and prevented HIF-1α induction via SDH inhibition, which restored Sirt2 activity by improving intracellular redox homeostasis. 

According to Alexander et al. [62], administering the SDH inhibitor dimethyl malonate (DDM) in myeloid-specific Bmal1 knockout (M-BKO) macrophages could restore a several-fold induction of HIF-1α protein levels, resulting from activation by M1 and suppression of tumor growth. Ruan et al. [63] investigated the way in which paeoniflorin (PF) intriguingly aggregated SDH activity to dampen the high expression of HIF-1α, thus, alleviating inflammatory pain in a peripheral inflammatory pain model. Hence, these findings explain the cause of highly vascular tumors that develop without VHL mutations, as affected by mutations in SDH. FH is also associated with HIF-1α stabilization and activity. Similarly, FH loss leads to the accumulation of fumarate, which can competitively inhibit prolyl hydroxylase that hydroxylate HIF-1α and HIF- 2α. Apart from the direct inhibition of prolyl hydroxylase by fumarate, it was also shown that FH inactivation increased cellular ROS, which in turn drove constitutive HIF activation [64]. Tseng et al. [65] demonstrated that the depletion of the transketolase (TKT) or the addition of alpha-ketoglutarate (aKG) enhanced the levels of the tumor suppressors FH and SDH, and decreasd the oncometabolites succinate and fumarate. Those results led to further stabilization of PHD2 and the inhibition of HIF-1α in a model of breast cancer cells, ultimately suppressing breast cancer metastasis.

The matrix of mitochondria also contains thioredoxin (Trx2) and thioredoxin reductase (TrxR2). In addition to maintaining the stability of cytoplasmic proteins, they can also act on the build-up of HIF-1α and the stimulation of its transcription. Zhou et al. [66] presented evidence that the overexpression of Trx2 or TrxR2 attenuated NO-evoked HIF-1α accumulation and the transactivation of HIF-1 in HEK293 cells. Since the thioredoxins interfered with Akt/mTOR signaling, as well as p42/44 mitogen-activated protein kinase and p70S6 kinase functions, studies indicated that these activities were under the control of thioredoxins. 

### 5.2. Mitochondrial Respiratory Chain and HIF-1 Stability 

Most studies believe that all complexes of the respiratory chain are involved in the regulation of the stability of HIF-1α, which might be achieved indirectly through oxygen consumption. According to Agani et al. [67], the neurotoxin 1-methyl-4-phenyl-1,2,3,6-tetrahydropyridine (MPTP) inhibited complex I activity, as well as the accumulation of HIF-1α protein in the striatum of C57BL/6 mice following a hypoxic challenge. It has been demonstrated that succinate, which is part of the mitochondrial complex II subunits, restores the hypoxia response in cybrid cells, suggesting that respiratory chain activity is required to activate HIF-1. Mateo et al. [68] regulated in tetracycline-inducible Tet-iNOS 293 cells, which indicated that low concentrations of NO (<400 nM) cause a rapid decrease in HIF-1α due to the inhibition of the mitochondrial respiratory chain. They also showed similar results when using other respiratory chain blockers, such as sodium azide, antimycin, and rotenone; however, high levels of NO in the atmosphere (>1 μM) maintained HIF-1α expression independent of mitochondria. Indirect measurements of the intracellular oxygen concentration demonstrate that the intracellular oxygen concentration is higher when the respiratory chain is blocked. By blocking the respiratory chain, the oxygen consumption of the respiratory chain is reduced and the oxygen concentration in the cellular microenvironment is increased, degrading HIF-1α. Moreover, many studies believe that ROS plays an indispensable role in the stabilization of HIF-1, and ROS production is mainly controlled by the respiratory chain complex. Therefore, it has been said that the initiation of HIF-1 activity comes from the respiratory chain complex. Using RNA interference of SdhB or the pharmacological inhibition of distal subunits of complex II, Guzy et al. [70] demonstrated that HIF-α stabilization was increased in an ROS-dependent manner when the distal subunits of complex II were inhibited. Moreover, suppressing HIF-1α and/or HIF-2α inhibited the enhanced growth rates of tumor cells, resulting from SdhB suppression, suggesting that ROS production and the subsequent activation of HIF-α were responsible for SdhB-induced tumor formation. Chandel et al. [71,72] confirmed that hypoxia increased mitochondrial ROS generation at Complex III, which led to the collection of HIF-1α protein, revealing that mitochondria-derived ROS were both required and sufficient to initiate HIF-1α stabilization during hypoxia. That hypothesis has also been supported by other studies, which demonstrates the regulation of HIF activity by mitochondrial-derived ROS under hypoxia, growth factors [73,74,75], and antioxidants [76,77,78]. Thus, the mechanism might involve phosphatidylinositol 3-kinase (PI3K)-Akt / protein kinase C (PKC) / histone deacetylase (HDAC) and p38 mitogen-activated protein kinase signaling [79,80].

However, these findings are controversial on the effects of the mitochondrial respiratory chain on HIF-1, and there are also some negative reports. Gong et al. [81] demonstrated that various tumor cell lines were suppressed by the inhibition of electron transport complexes I, III, and IV, as well as by inhibition of mitochondrial F0F1 ATPase. The findings suggested that the impaired oxygen consumption and inhibition of the electron transport chain affected HIF-1α accumulation in hypoxic cells. Tan et al. [82] found that, in older myocytes, HIF-1 increased inotropy without affecting mitochondrial activity, with no change in the activities of Complexes I and III. Tuttle et al. [83] did not find any evidence to suggest that ROS, produced by mitochondria, were necessary for stabilizing HIF-1α under moderate hypoxia. Under moderate and severe hypoxia, the oxygen dependence of the prolyl hydroxylase reaction is sufficient to mediate HIF-1α stability. Hagen et al. [84] found that anti-oxidants, such as ascorbate, glutathione, and N-acetylcysteine, did not reverse the inhibition of hypoxia-dependent HIF-1α stabilization by myxothiazol. Furthermore, antioxidants did not affect hypoxia-dependent HIF-1α stabilization. This might be due to the difference of cell types or the concentrations of hypoxia, and then mitochondrial function could be affected by the metabolic status. 

### 5.3. Uncoupling Proteins and HIF-1 Activity

Uncoupling proteins (UCPs) are located in the IMM and are a superfamily of mitochondrial anion carrier proteins. They are extremely sensitive to O_2_ and play an important role in reducing MMP and inhibiting ROS production, which can reduce lipid peroxidation to a certain extent and protect cells. However, due to uncoupling, the synthesis of intracellular ATP is reduced, and sufficient ATP cannot be obtained when the cell is in urgent need of energy.

Studies have found that UCPs affect the stability and activity of HIF-1. In the study of PC-3 and DU-145 prostate cancer cells, Thomas et al. [85] demonstrated that mitochondrial uncouplers, rottlerin and FCCP, tremendously reduced hypoxic and normoxic HIF-1 transcriptional activity; the process was partly mediated by a reduction in the levels of oxygen labile HIF-1α and HIF-2α proteins. In addition, mitochondrial uncouplers reduce the expression of HIF target genes, VEGF, and VEGF receptor-2. According to the study, mitochondrial uncouplers are likely to be effective in inhibiting HIF pathway activity in tumors. Ke et al. [86] showed that in UCP2-deficient proximal tubular cells (PTCs), the inhibition of tubulointerstitial fibrosis (TIF) resulted from the downregulation of HIF-1α. The results suggested that UCP2 regulated TIF by stimulating the HIF-1α stabilization pathway in tubular cells, establishing it as a possible therapeutic target for the treatment of chronic renal fibrosis. Using the fibroblasts cells from an MTO1 patient, Boutoual et al. [87] indicated that MTO1 fibroblasts down-regulated PPARγ, UCP2, SREBP-1c and PPARβ/δ, exhibiting activation of HIF-1 and inactivation of AMPK, leading to the reprogramming of the cellular metabolism mediated by the HIF-PPARγ-UCP2-AMPK axis and the contributions of PPARβ/δ and SREBP-1c.

### 5.4. Mitochondrial ATP-Sensitive Potassium Channels and HIF-1 Activity

Mitochondria ATP-sensitive potassium channels (Mito K_ATP_) are important structures that couple cellular energy metabolism with cellular electrical activity, thereby affecting cellular functional activities. 

Several previous studies have suggested that the Mito K_ATP_ channel activation promotes HIF-1 expression. The results of Yilmaz et al. [88] indicated that EPO inhibited K_ATP_ channels in human renal tubular cells CRL-2830 under hypoxic/normal conditions, suggesting that EPO might have cytoprotective effects through these channels. By means of the rat heart perfusion model, Li et al. [89] speculated that ischemic post-conditioning (IPO) and diazoxide post-conditioning (DPO) might open Mito K_ATP_ channels, and, in turn, activated the HIF-1/Hre pathway to alleviate myocardial ischemia reperfusion injury (Miri).

## 6. Conclusions

At present, there have been systematic and deep studies on both the structural changes and functional regulations of mitochondria by HIF-1, as well as the initiating role of mitochondria on the stability and activity of HIF-1. Many meaningful signs of progress have been achieved.

In summary, HIF-1 mediates adaptation to hypoxic conditions by reducing mitochondrial activity, thereby reducing cellular oxygen dependence. In addition, mitochondria are also involved in HIF signaling by transmitting a large amount of metabolic stress to the HIF pathway.

As a vital part of energy metabolism, mitochondria are the energy factories of cells. Their oxygen-dependent metabolism is very important for the body to maintain normal life activities, and their structure and function are particularly important. CREB regulates mitochondrial function. Mitochondrial DNA damage during hypoxia causes the altered expression of several genes, including HIF-1α. The exact mechanism of action remains to be explored. The role of HIF-1 in modulating mtDNA and mitochondrial functions, such as mitochondrial genesis and mitochondrial autophagy, via altering CREB expression deserves further investigation.

The core role of HIF-1 has been gradually revealed in the study of hypoxia. It makes HIF-1 a central regulator of hypoxia and other stimuli by the extensive pathophysiological effects of HIF-1 and numbers of target genes. Meanwhile, HIF-1 can also be stimulated by a variety of extracellular factors and plays a pivotal role in cell survival, growth, and differentiation. The interaction and connection between HIF-1 and mitochondria can combine their powerful biological functions, opening a new page for the research on the mechanisms and prevention of various injuries.

However, the potential relationship between HIF-1 and mitochondria remains to be further explored. What are the molecular mechanisms involved in the mitochondrial genome-mediated stable expression of HIF-1 protein? What is involved in the activation of HIF-1 and the relating signal transcription pathways? The synthesis of HIF-1 depends on the activity and expression of its α subunit. What role do mitochondria play in the synthesis process, and what is the specific mechanism? Further related research can deepen the understanding of the interaction mechanism between HIF-1 and mitochondria under different conditions, will provide new targets and ideas for the development of various drugs and disease treatments, continue research on energy metabolism and hypoxia adaptation. To provide a scientific basis, this study is significant and highly valuable.

## Figures and Tables

**Figure 1 biomolecules-13-00050-f001:**
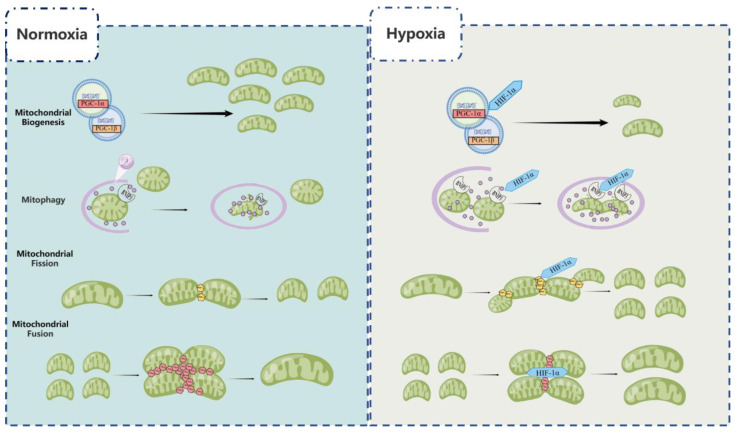
Regulation of mitochondrial number by HIF-1.

**Figure 2 biomolecules-13-00050-f002:**
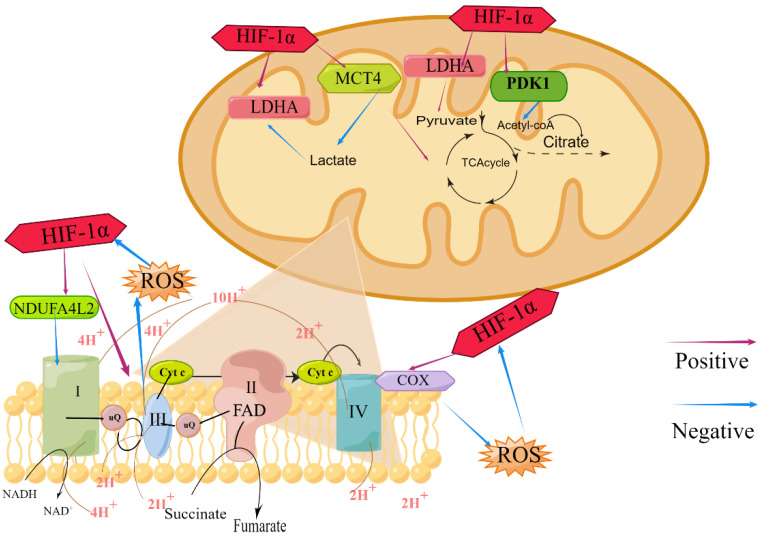
The effects of HIF-1 on TCA cycle, electron respiratory chain and ROS production in mitochondrial function.

**Table 1 biomolecules-13-00050-t001:** The effect of mitochondria on HIF-1 activity or stability.

Related Mitochondrial Function	Proteins or Receptors	Effects on HIF-1	Ref. No.
Mitochondrial metabolic enzymes	Succinate dehydrogenase(SDH) Fumarate hydratase(FH)	Negative	[59,60,61,62,63,64,65]
	Thioredoxin (Trx2) and Thioredoxin reductase (TrxR2)	Positive	[66]
Mitochondrial respiratory chain	Complex I II III IV	Positive	[67,68,69]
	ROS	Positive/Negative	[70,71,72,73,74,75,76,77,78,79,80,81,82,83,84]
Uncoupling proteins	Rottlerin and franchise control chain project(FCCP)	Negative	[85]
	UCP2	Negative	[86,87]
Mitochondrial ATP-sensitive potassium channels	Mito K_ATP_ channel	Positive	[88,89]

## Data Availability

Not applicable.

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
