# Peer review of "Hypoxia-Inducible Factor 1 and Mitochondria: An Intimate Connection"

_biomolecules, 2022, doi:10.3390/biom13010050_

Round 1

Reviewer 1 Report

The present review titled “Hypoxia-inducible factor 1 and mitochondria: an intimate connection” is focused on interaction and connection between HIF-1 and mitochondria. The interrelationships between HIF-1 and mitochondria represent interesting targets for developing new drugs and  treatment that involve energy metabolism and hypoxia adaptation.

 The reference list is old and I suggest its updating.

Overall, the manuscript is well written, even if some minor stylistic revisions should be made.

I think this review is acceptable for publication in Biomolecules.

Reviewer 2 Report

This is a very detailed (sometimes too detailed) review of a complex issue that tries to highlight the interrelation between hypoxia, the hypoxia-responsive element HIF-1, and mitochondria function in the cell. The review is generally written well, touching on all the important issues in the mutual interactions between mitochondria and HIF-1. Although the review concentrates on HIF-1, I would have expected that in the introduction and conclusion sections,  the mention of an additional pathway controlled by CREB that was shown to be a hypoxia-responsive regulator, plays an important role in the expression of genes in the mitochondria.

specific comments

line 43:  "The HIF complex is abundant when oxygen is present"  is this true?

line 50: the word "exactly " should not be used here

lines 80-82: are repeated in lines 86-88

lines 83-84: please clearify.

lines 98-99: is the number of mitochondria stable or as written in the same sentence " the number will change with the fluctuation of cell function"?

4.1.2 please explain the differences in mitochondria function in the cell under anterograde movement and retrograde.

Reviewer 3 Report

review is very good, but i have two comments:

1. can you explain in details the role of iron in mitochonria during hypoxia

2. The review needs some further recent  references
